# Origins of the 2009 H1N1 influenza pandemic in swine in Mexico

Ignacio Mena[1,2†], Martha I Nelson[3†], Francisco Quezada-Monroy[4], Jayeeta Dutta[5], Refugio Cortes-Fernández[4], J Horacio Lara-Puente[4], Felipa Castro-Peralta[4], Luis F Cunha[5], Nídia S Trovão[1,2,3,6], Bernardo Lozano-Dubernard[4], Andrew Rambaut[3,7,8], Harm van Bakel[5], Adolfo García-Sastre[1,2,9*]

[1]Department of Microbiology, Icahn School of Medicine at Mount Sinai, New York, United States; [2]Global Health and Emerging Pathogens Institute, Icahn School of Medicine at Mount Sinai, New York, United States; [3]Division of International Epidemiology and Population Studies, Fogarty International Center, National Institutes of Health, Bethesda, United States; [4]Laboratorio Avi-Mex, Mexico City, Mexico; [5]Genetics and Genomic Sciences, Icahn School of Medicine at Mount Sinai, New York, United States; [6]Department of Microbiology and Immunology, Rega Institute, University of Leuven, Leuven, Belgium; [7]Institute of Evolutionary Biology, University of Edinburgh, Edinburgh, United Kingdom; [8]Centre for Immunology, Infection and Evolution, University of Edinburgh, Edinburgh, United Kingdom; [9]Department of Medicine, Division of Infectious Diseases, Icahn School of Medicine at Mount Sinai, New York, United States

*For correspondence: adolfo.
garcia-sastre@mssm.edu

†These authors contributed
equally to this work

Competing interests: The
authors declare that no
competing interests exist.

Reviewing editor: Richard A
Neher, Max Planck Institute for
Developmental Biology,
Germany

**Abstract** Asia is considered an important source of influenza A virus (IAV) pandemics, owing to large, diverse viral reservoirs in poultry and swine. However, the zoonotic origins of the 2009 A/H1N1 influenza pandemic virus (pdmH1N1) remain unclear, due to conflicting evidence from swine and humans. There is strong evidence that the first human outbreak of pdmH1N1 occurred in Mexico in early 2009. However, no related swine viruses have been detected in Mexico or any part of the Americas, and to date the most closely related ancestor viruses were identified in Asian swine. Here, we use 58 new whole-genome sequences from IAVs collected in Mexican swine to establish that the swine virus responsible for the 2009 pandemic evolved in central Mexico. This finding highlights how the 2009 pandemic arose from a region not considered a pandemic risk, owing to an expansion of IAV diversity in swine resulting from long-distance live swine trade.

## Introduction

Our ability to predict outbreaks of zoonotic pathogens requires an understanding of their ecology and evolution in reservoir hosts. At the onset of the 2009 influenza pandemic, whole-genome sequencing revealed that pdmH1N1 was a novel reassortant virus comprised of segments from three major swIAV lineages (*Garten et al., 2009*): segments PB2, PB1, PA, NP and NS were derived from a triple reassortant H3N2 swine virus (TRsw) that originated in North American swine during the mid-1990s; the HA (H1) segment derived from the classical swine H1N1 lineage (Csw) that has circulated in North American swine since the 1918 pandemic; and the NA (N1) and MP segments were related to the avian-like Eurasian swine lineage (EAsw) that emerged in European pigs in the late 1970s. Genetic and epidemiological evidence indicate that the first outbreak of pdmH1N1 occurred in humans in Mexico (*Brockmann and Helbing, 2013*; *Chowell et al., 2011*; *Lemey et al., 2009*). However, it remained unclear whether Mexico also was the site of the zoonotic transmission event

**eLife digest** In 2009 a new influenza virus jumped from pigs to humans and spread very rapidly, causing an initial outbreak in Mexico and becoming a global pandemic in just a few months. Although the most straightforward explanation is that the virus originated in swine in Mexico, several studies suggested that this was unlikely because key genetic components of the virus had never been detected in the Americas. Determining the source of the disease is critical for predicting and preparing for future influenza pandemics.

Mena, Nelson et al. sought to better characterize the genetic diversity of influenza viruses in Mexican swine by obtaining the entire genetic sequences of 58 viruses collected from swine in Mexico, including some from previously unsampled regions in central Mexico. The sequences revealed extensive diversity among the influenza viruses circulating in Mexican swine. Several viruses included genetic segments that originated from viruses from Eurasia (the landmass containing Europe and Asia) and had not previously been detected in the Americas. The new sequences contained key genetic components of the 2009 pandemic virus. Furthermore, the sequences suggest that viruses with a similar genetic composition to the 2009 pandemic virus have been circulating in pigs in central-west Mexico for more than a decade. Thus, this region is the most likely source of the virus that started the 2009 pandemic.

Mena, Nelson et al. also found that the movement of viruses from Eurasia and the United States into Mexico closely follows the direction of the global trade of live swine. This highlights the critical role that animal trading plays in bringing together diverse viruses from different continents, which can then combine and generate new pandemic viruses.

A potential next step is to perform experiments that investigate how well the swine viruses can replicate and pass between different animal models. Comparing the results of such experiments with the findings presented by Mena, Nelson et al. could identify factors that make the viruses more likely to spread to humans and produce a pandemic.

---

that gave rise to the 2009 outbreak, given that EAsw viruses had never been detected in swine in any part of the Americas, including Argentina, Brazil, Chile, Mexico, Peru, Canada, and the intensively sampled swine herds in the United States (*Anderson et al., 2013*; *Nelson et al., 2015b*; *Pereda et al., 2011*; *Tinoco et al., 2015*). At the time, Asia was the only region where TRsw, Csw, and EAsw viruses were known to co-circulate in swine (*Zhu et al., 2013*).

Asia is a global source of novel human seasonal influenza viruses (*Lemey et al., 2014*; *Russell et al., 2008*), avian viruses associated with the pandemics of 1957 and 1968 (*Kawaoka et al., 1989*), and emerging avian viruses with pandemic potential (*Lam et al., 2015*; *Taubenberger Morens, 2009*). Since 2009, several studies have implicated Asian swine as a possible source of the TRsw/Csw/EAsw reassortant swine virus that gave rise to pdmH1N1. Asia is the only region where TRsw, Csw, and EAsw are known to exchange segments through reassortment (*Lam et al., 2011*; *Poonsuk et al., 2013*; *Takemae et al., 2008*; *Vijaykrishna et al., 2011*). At the onset of the pandemic, IAVs identified in swine (swIAVs) in Hong Kong, SAR, were more closely related to pdmH1N1 than any other swIAVs available globally (*Smith et al., 2009*). One Hong Kong swIAV was identified with a genotype similar to the pdmH1N1 virus except for the NA segment, and transmitted via respiratory droplet in ferrets (*Yen et al., 2011*). However, the time of the most recent common ancestor (tMRCA) of the Hong Kong swIAVs and pdmH1N1 was still approximately 10 years (*Smith et al., 2009*). Combined with the lack of any complete swIAV genomes from Mexico's large swine populations at the start of the pandemic, and the difficulty of understanding how a virus that evolved in Asian swine caused its first outbreak in humans in Mexico, the geographical location of the swine-to-human transmission event that gave rise to pdmH1N1 has remained uncertain. In the years following the pandemic, new surveillance in Mexican swine has identified Csw, TRsw, and pdmH1N1 and seasonal H3N2 viruses of human origin (*Nelson et al., 2015a*), but no EAsw viruses. However, the majority of swIAVs in previous analyses were collected in northern and eastern Mexico, and no studies to date have included viruses from major swine-producing states in central Mexico, including Jalisco, Guanajuato, and Puebla (*Figure 1*).

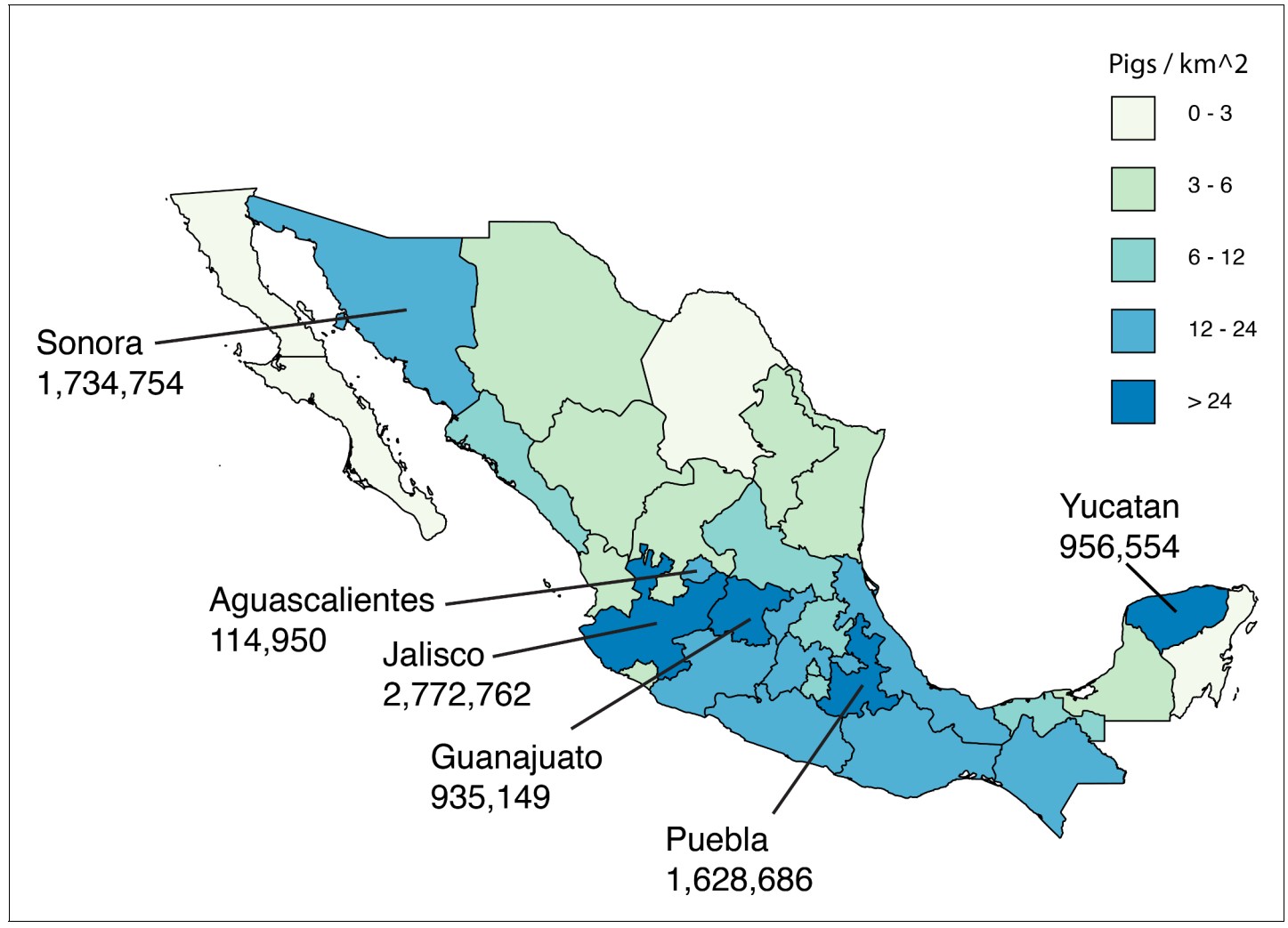

**Figure 1.** Live swine production in Mexico. Each Mexican state is shaded according to the density of pigs (the number of pigs per square kilometer, light green = lower and dark blue = higher). The six Mexican states where influenza viruses were collected for this study are indicated, and the number of pigs in the state is provided below the state's name. Source data from Mexico's Secretariat of Agriculture, Livestock, Rural Development, Fisheries and Food (SAGARPA) is available in *Figure 1—source data 1*.

The following source data is available for figure 1:

**Source data 1.** Swine population sizes in Mexican states in 2014.

Resolving whether the viral precursors of pdmH1N1 evolved in swine in Mexico or Asia has important implications for understanding pandemic emergence and informing risk assessment. We therefore undertook expansive surveillance efforts in Mexico, isolating the virus from pigs with respiratory symptoms in farms from six Mexican states with high swine production, including Sonora in northern Mexico, Yucatan in eastern Mexico, and previously unsampled states in central-east Mexico (Puebla), and central-west Mexico (Jalisco, Guanajuato, and Aguascalientes, *Figure 1*). These efforts yielded 58 complete viral genomes (sequence accession numbers are available in the Dryad data repository under Doi:10.5061/dryad.m550m) that were used to identify swIAVs that are closely related to pdmH1N1, resolving the origins of pdmH1N1 in its reservoir host.

## Results

### Extensive IAV genetic diversity identified in Mexican swine

Among the 58 swIAVs newly sequenced from Mexico, genome segments were identified from all three major swIAV lineages (Csw, TRsw, and EAsw) and all three major human IAV lineages of the 21st century (H3N2, pdmH1N1, and pre-pandemic H1N1), evidence of extensive gene flow between Mexican swine and (a) humans, (b) US swine (Csw and TRsw viruses), and (c) European swine (EAsw viruses) (*Figure 2*). Except for TRsw (genotype 5) and pdmH1N1 (genotype 6), no other IAV lineages were identified intact in Mexican swine, evidence of frequent reassortment between viral lineages. Of the 12 genotypes identified, 10 were reassortants. Four genotypes were triple reassortants: genotype 1 contains all three swine lineages (Csw/TRsw/EAsw); genotype 9 contains all three human-origin lineages (pre-pandemic H1N1/human H3N2/pdmH1N1); and genotypes 2 and 11 contain one human-origin and two swine lineages (pre-pandemic H1N1/TRsw/EAsw and pdmH1N1/ Csw/TRsw, respectively). Although human-origin viruses were observed in all regions of Mexico, the importance of human-to-swine transmission in the evolution of swIAV diversity in Mexico has been characterized recently in detail (*Nelson et al., 2015a*). Instead, this study will focus on evolutionary processes that are directly involved in the emergence of pdmH1N1 precursors in Mexican swine, including reassortment and viral migration driven by long-distance movements of live swine.

### Independent evolution of swIAVs in regions of Mexico

A high degree of population structure was observed among swIAV populations in Mexico's northern, central, and eastern regions. Viruses collected in Sonora, Yucatan, Puebla, and Jalisco/Guanajuato/ Aguascalientes consistently were positioned in different sections of the phylogenetic trees inferred for individual genome segments (*Figure 3*), evidence of independent viral introductions into these regions from humans, US swine, and Eurasian swine (summarized in *Figure 4A*). Viral gene flow was not frequently observed between Mexico's northern, eastern, and central regions (*Figures 3*,*4A*), a contrast to the frequency of inter-regional IAV migration observed in US swine herds, driven by ongoing long-distance movement of animals (*Nelson et al., 2011*). The only evidence for a migration event between two Mexican regions that is observed for multiple segments involves a single Yucatan virus (AVX-57) that is closely related to Jalisco viruses, evidence of Jalisco-to-Yucatan migration (*Figure 4A*, *Figure 3—source data 1*). Viral migration occurred frequently between states in Mexico's central-west region, as evidenced by the clustering of viruses from Jalisco, Guanajuato, and Aguascalientes within the same phylogenetic clades (*Figure 3—source data 1*). Jalisco's large swine herds were found to be an important source of viruses in the neighboring states of Guanajuato and Aguascalientes (*Figure 4A*, *Figure 4—figure supplement 1*).

### Multiple introductions of swIAVs from US and European swine into Mexico

Time-scaled phylogenies suggest that viral introductions from US and European swine into Mexico have occurred periodically for at least two decades, coinciding with a sharp increase in the number of reported imports of live swine into Mexico in the late 1980s (*Figure 5*). Notably, EAsw viruses have been introduced from Eurasian swine into central Mexico on at least two occasions. Although the US and Canada are Mexico's primary trade partners for the import of live swine, live swine imports into Mexico also were reported from several European countries during the late 1990s, including the United Kingdom and Denmark (*Figure 6*). We estimate that at least three TRsw viruses and two Csw viruses were introduced from US swine into Mexican swine herds in the northern and central regions. Estimates of the number and timing of independent swIAV introductions into Mexico are complicated by gaps in global background swIAV sequence data in the 1990s/early 2000s and topological differences in the phylogenies inferred for different segments arising from reassortment. Our conservative approach of defining introductions based on both high posterior probabilities (>90) on the phylogeny (*Figure 3—source data 1*) and Bayes factor (BF) support for significant rates of migration (BF > 6) (*Figure 4—figure supplement 1*) has likely underestimated the number of introductions. We also recognize the need to draw phylogeographic inferences with an awareness of missing data. Major gaps in surveillance in swine in Europe and Asia during the 1990s have resulted in long branch lengths between swIAVs collected in Mexico and Eurasia (*Figure 3—source data 1*).

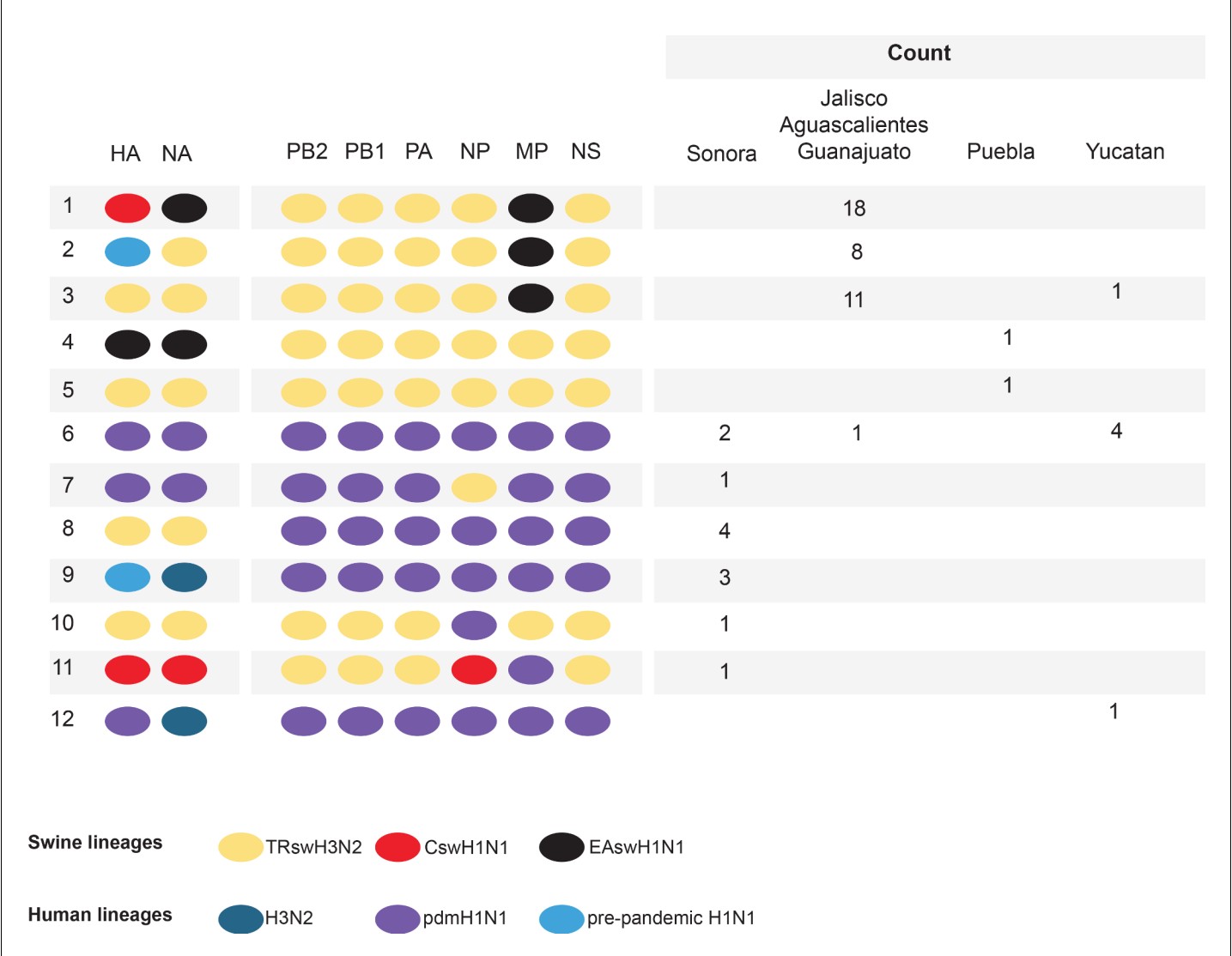

**Figure 2.** Genetic diversity of IAVs in Mexican swine, 2010–2014. Twelve genotypes were identified by surveillance in Mexican swien herds during 2010–2014. Each oval represents one of the eight segments of the viral genome. The surface antigens HA and NA are listed first, followed by the six internal gene segments. The shading of each oval corresonds to the genetic lineage of IAVs found in humans and swine globally. The number of swIAVs with a given genotype is indicated for each region in Mexico : Sonora (northern Mexico), Jalisco/Aguascalientes/Guanajuato (central-west Mexico), Puebla (central-east Mexico), and Yucatan (eastern Mexico). The genotype and additional characteristics of each of the 58 swIAVs collected and sequenced from Mexico for this study are provided in *Figure 2—source data 1*.
The following source data is available for figure 2:

**Source data 1.** Characteristics of the 58 svIAVs collected in Mexico for this study.

Although it is difficult to infer from the tree whether the EAsw MP identified in Mexico was introduced from Asian or European swine (*Figure 4—figure supplement 1*), the lack of live swine imports reported between Mexico and any Asian country (*Figure 6*), and the low export of Asian swine globally (*Nelson et al., 2015c*) suggest that Europe may be a more likely source of the Mexican viruses, similar to the H1 and N1 segments.

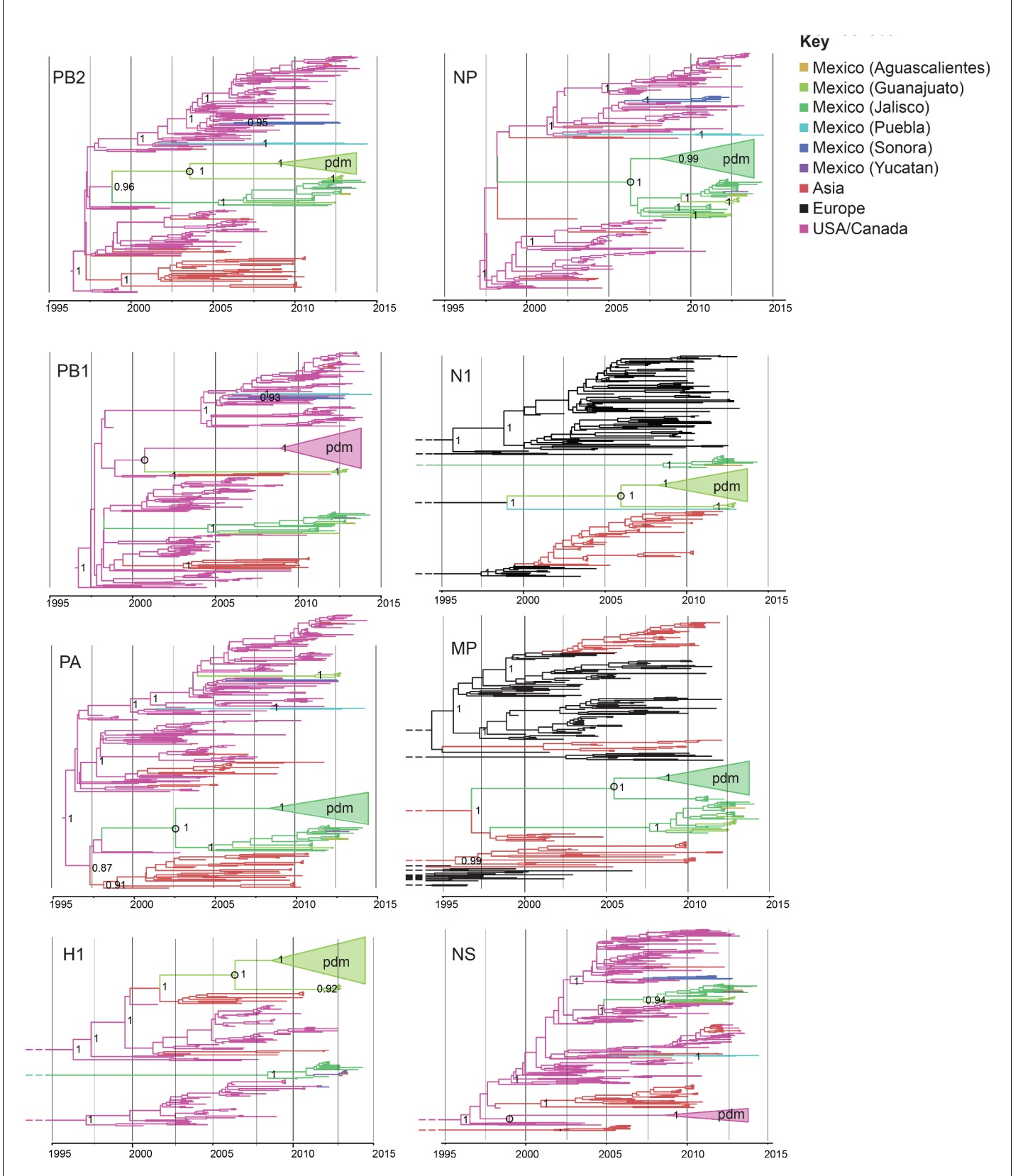

**Figure 3.** Evolutionary relationships between swIAVs collected in Mexico and pdmH1N1. Time-scaled Bayesian MCC trees inferred for the eight segments of the IAV genome, for the lineages found in pdmH1N1: TRIG (PB2, PB1, PA, NP, and NS), classical (H1), and avian-like Eurasian (N1 and MP). *Figure 3 continued on next page*

*Figure 3 continued*

Trees include the 58 swIAVs collected in Mexico for this study, representative pdmH1N1 viruses, and other related swIAVs collected globally. The color of each branch indicates the most probable location state. For clarity, the pdmH1N1 clade is depicted as a triangle, the color of which represents the inferred location state of the node representing the inferred common ancestor of pdmH1N1 and the most closely related swIAVs (indicated by an open circle). Posterior probabilities are provided for key nodes. More detailed phylogenies including tip labels are provided in *Figure 3—source data 1*. along with trees inferred for lineages not shown here. Similar phylogenies inferred using maximum likelihood methods are provided in *Figure 3—source data 2*. Similar phylogenies that use genotype instead of geographic location as a trait are provided in *Figure 3—source data 3*. Nine Mexican swIAVs were excluded from the phylogenetic analysis because they were outliers in root-to-tip divergence (*Figure 3—figure supplement 1*). More detailed phylogenies of the pdmH1N1 viruses reveal multiple independent introductions from humans into swine in Mexico (*Figure 3—source data 4*).

The following source data and figure supplement are available for figure 3:

**Source data 1.** MCC trees presenting the evolutionary relationships between swIAVs collected in Mexico and swIAVs and human IAVs collected globally, for each IAV segment as well as for each IAV lineage found in Mexican swine: PB2 (TRIG/pdmH1N1), PB1 (TRIG/pdmH1N1), PA (TRIG/pdmH1N1), H1 (classical/pdmH1N1), H1 (avian-like Eurasian), H1 (human seasonal/human- like swine), H3 (human seasonal/human-like swine), NP (TRIG/classical/pdmH1N1), N1 (classical), N1 (avian-like Eurasian/pdmH1N1), N2 (human seasonal/human-like swine), MP (TRIG/classical), MP (avian-like Eurasian/pdmH1N1), NS (TRIG/classical/pdmH1N1).
**Source data 2.** Maximum likelihood trees with tip labels.
**Source data 3.** Time spent in a genotype using Markov rewards.
**Source data 4.** Detailed phylogenetic analysis of pdmH1N1 in Mexican swine.
**Figure supplement 1.** Mexican swIAVs excluded from the phylogenetic analysis.

## Identification of pdmH1N1 precursor viruses in central Mexican swine

An important consequence of the long-distance dissemination of EAsw, Csw, and TRsw viruses into central Mexico was the identification of 18 swIAVs in central-west Mexico with the same combination of EAsw, Csw, and TRsw segments as pdmH1N1 (referred to as genotype 1, *Figure 2*). Genotype 1 viruses are distinct from the reverse zoonosis introductions of pdmH1N1 from humans to swine (genotype 6) that have been observed globally (*Nelson and Vincent, 2015*). Phylogenetically, genotype 1 viruses from Mexico form a sister lineage to pdmH1N1 on the PB2, PB1, PA, HA, NP, and NA trees (*Figure 3*, *Figure 3—source data 2*). Genotype 2 viruses from Jalisco (genotype 2 viruses are similar to genotype 1, but contain a pre-pandemic human-origin H1 segment [*Figure 2*]) are closely related to pdmH1N1 on the MP tree. No Mexican swIAVs were closely related to pdmH1N1 on the NS tree. However, NS is the segment with the highest tMRCA between pdmH1N1 and the most closely related swIAVs (*Figure 7*), an indication of larger gaps in sampling and missing data. It is therefore likely that the NS precursor in Mexico was not detected by surveillance or has been replaced in intervening years by reassortment. The tMRCA analysis therefore provides further indication that Mexico is the likely origin of the swine virus that gave rise to pdmH1N1. However, it is difficult to determine the precise location within central-west Mexico of zoonotic transmission, owing to high rates of gene flow and reassortment between swIAVs in Jalisco and Guanajuato and resulting differences in tree topologies (*Figures 3*,*4A*). Swine viruses from Guanajuato are most closely related to pdmH1N1 on the phylogenies inferred for the PB2, PB1, H1, and N1 segments. Jalisco viruses are most closely related to pdmH1N1 on the PA and MP trees. Both Guanajuato and Jalisco viruses are related to pdmH1N1 on the NP tree. Therefore, the swine-to-human transmission event could have occurred in Guanajuato, Jalisco, or possibly another state in central Mexico that has not been sampled.

## Patterns of reassortment in central Mexican swine

Of the 10 reassortant genotypes identified in Mexican swine, four genotypes were identified in central Mexican swine that contain at least one EAsw segment, indicative of multiple reassortment events involving EAsw viruses. Genotype 1 viruses emerged multiple times in central Mexico and sustained transmission longer than any other genotype in central Mexican swine, as estimated using 'Markov rewards' (*Figure 3—source data 3*). No EAsw PB2, PB1, PA, NP, and NS segments were

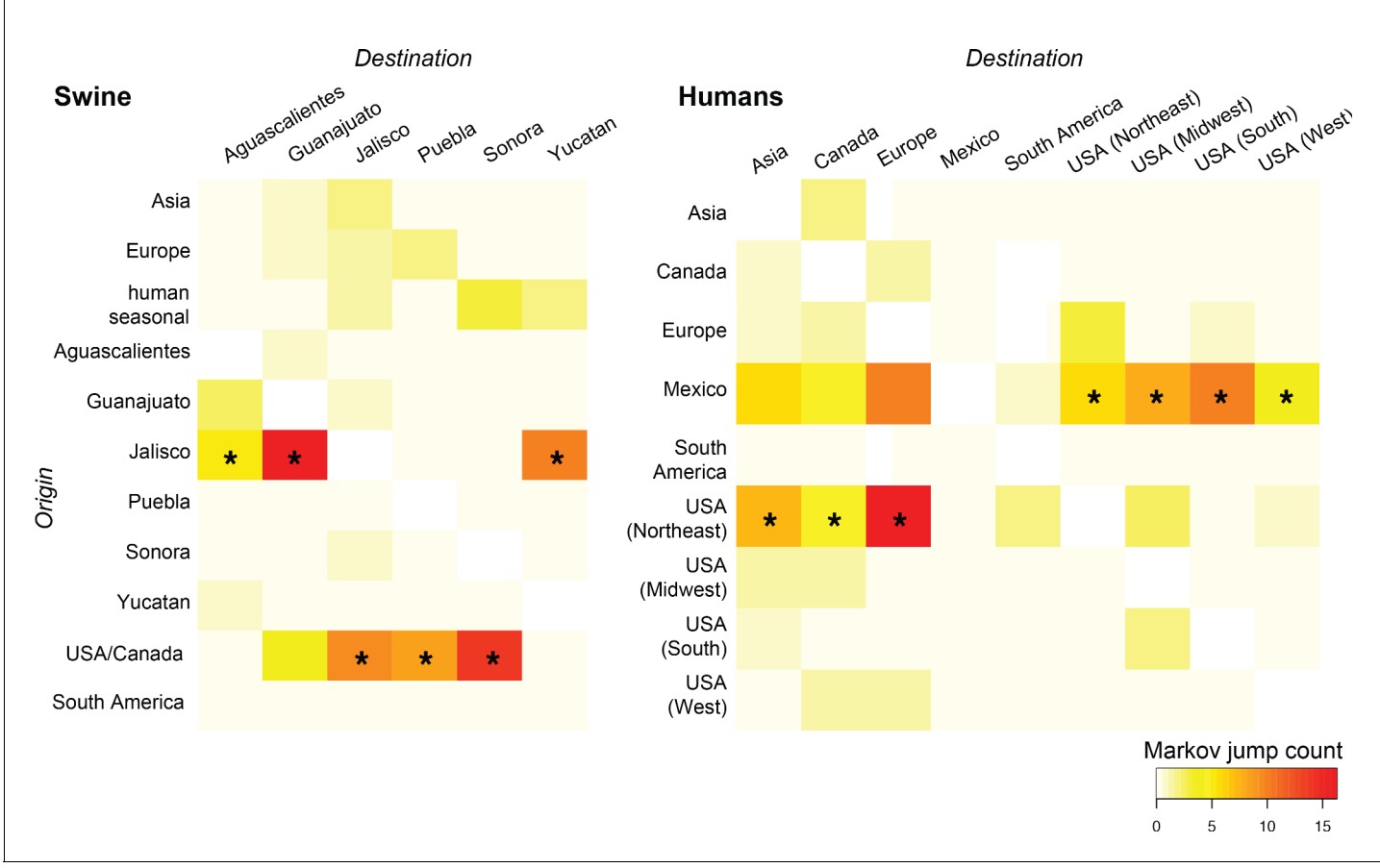

**Figure 4.** Heat-maps of IAV gene flow between locations. 'Markov jump' counts measure the number of inferred transitions, modeled by a continuous-time Markov chain process, that occur along the branches of the phylogeny, providing a measure of gene flow. The intensity of the color (red = high; white = low) reflects the number of Markov jump counts from a swine population in a location of origin (y-axis) to swine in one of six Mexican states (destination, x-axis; asymmetrical, summarized across all lineages and segments), and between human populations in Mexico, the United States, and globally during the early spatial dissemination of pdmH1N1 in humans during March–May 2009. Asterisks indicate the geographical source (y-axis) of the highest number of Markov jump counts for a particular destination (x-axis). Similar spatial linkages were observed using a Bayes factor (BF) test (*Figure 4—figure supplement 1*). A phylogenetic tree depicting the evolutionary relationships between human pdmH1N1 viruses is provided in *Figure 4—figure supplement 2*. Source data for both heat-maps is provided in *Figure 4—source data 1*.

The following source data and figure supplements are available for figure 4:

**Source data 1.** Expected number of location state transitions ('Markov jump' counts) along the branches of inferred phylogenies, summarized for (**a**) all segments and lineages identified in Mexican swine and (**b**) human pdmH1N1.

**Figure supplement 1.** Supported rates of viral migration.

**Figure supplement 2.** Phylogeography of pdmH1N1 in humans.

detected in any swine in Mexico, suggesting replacement by reassortment with Csw and TRsw viruses in central Mexico. One Puebla virus (AVX-47, genotype 4) was identified with a HA (H1) from EAsw. The tMRCA of this singleton virus and EAsw viruses from Europe is estimated to be in the early 1990s (*Figure 3—source data 1*). Although gaps in surveillance in European swine in the 1990s result in uncertainty about the precise timing of viral introduction from Europe-to-Mexico, it is possible that the EAsw H1 has circulated in Mexican swine undetected for many years, either in Puebla or another under-sampled state (*Figure 3—source data 1*). Notably, EAsw NA and MP segments were identified frequently in central Mexican swine. A conserved internal gene constellation of TRIG PB2,

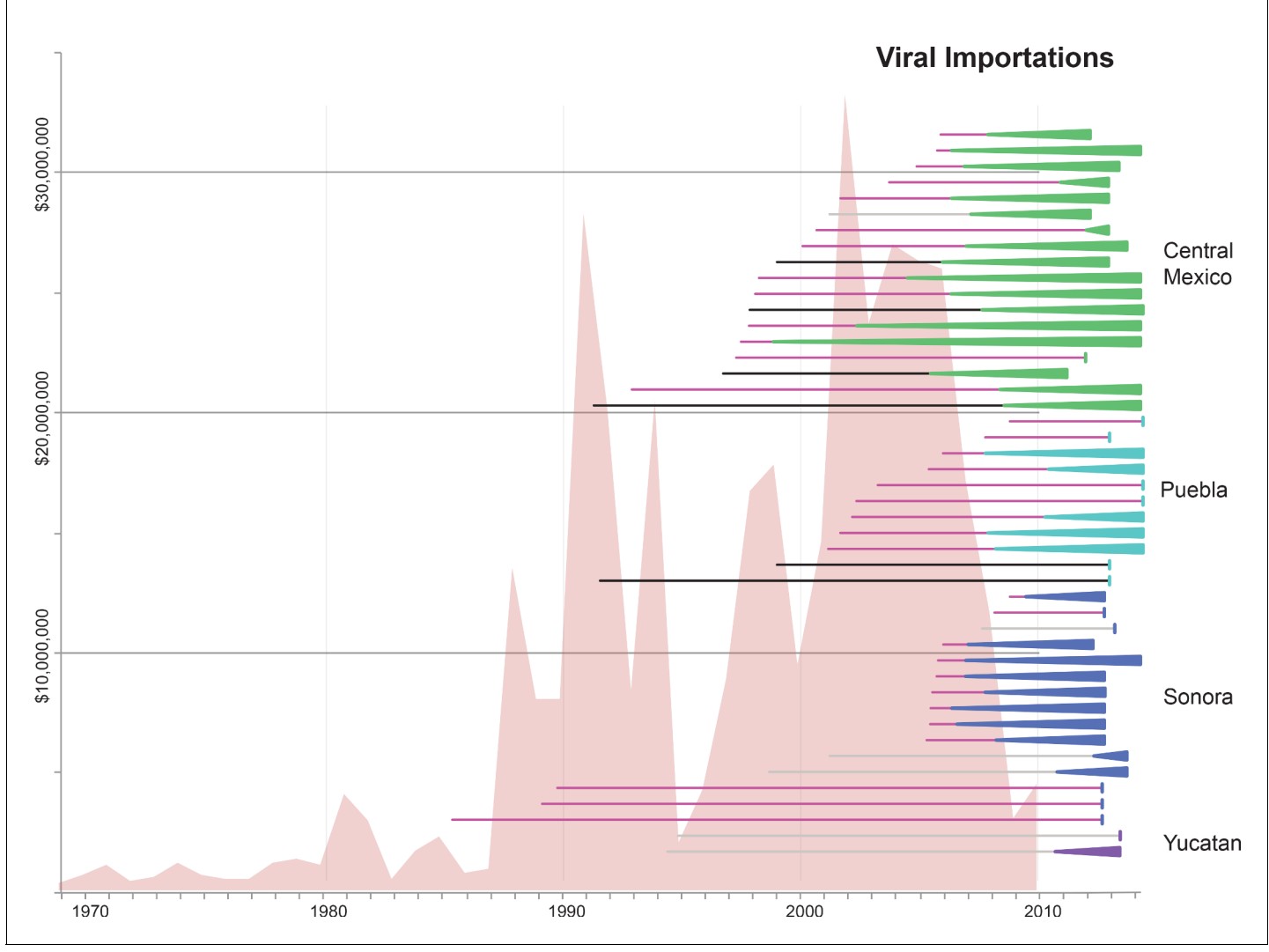

**Figure 5.** Import of live swine and IAVs into Mexico. The value of live swine imported into Mexico (USD, y-axis, left) from all countries during 1969–2010 is presented in the background in pink, based on trade data reported to the Food and Agricultural Organization (FAO) of the United Nationals, available in *Figure 5—source data 1*. Each horizontal line represents an introduction of an IAV segment into Mexican swine, the timing of which is inferred from the MCC trees. The shading of each line indicates the inferred location of origin of an introduction, consistent with *Figure 3*: dark pink = USA/Canadian swine, black = Eurasian swine, grey = humans. The length of the line indicates uncertainty in the timing of an introduction. Triangles represent clades resulting from onward transmission in Mexico and extend forward as far as the most recently sampled virus. The shading of each triangle indicates the destination location, consistent with *Figure 3*: dark purple = Yucatan, dark blue = Sonora, light blue = Puebla, and green = Jalisco, Aguascalientes, and Guanajuato (central-west Mexico). Lines without triangles represent singletons. A similar figure annotated with the segment associated with each introduction is provided in *Figure 5—figure supplement 1*.

The following source data and figure supplement are available for figure 5:

**Source data 1.** Reported value of live swine imports into Mexico from all countries during 1969–2010.

**Figure supplement 1.** Similar to *Figure 5*, but annotated with the segment associated with each introduction.

PB1, PA, NP, and NS segments and an EAsw MP segment was observed in over 97% of the swIAVs in Jalisco, Guanajuato, and Aguascalientes (*Figure 2*).

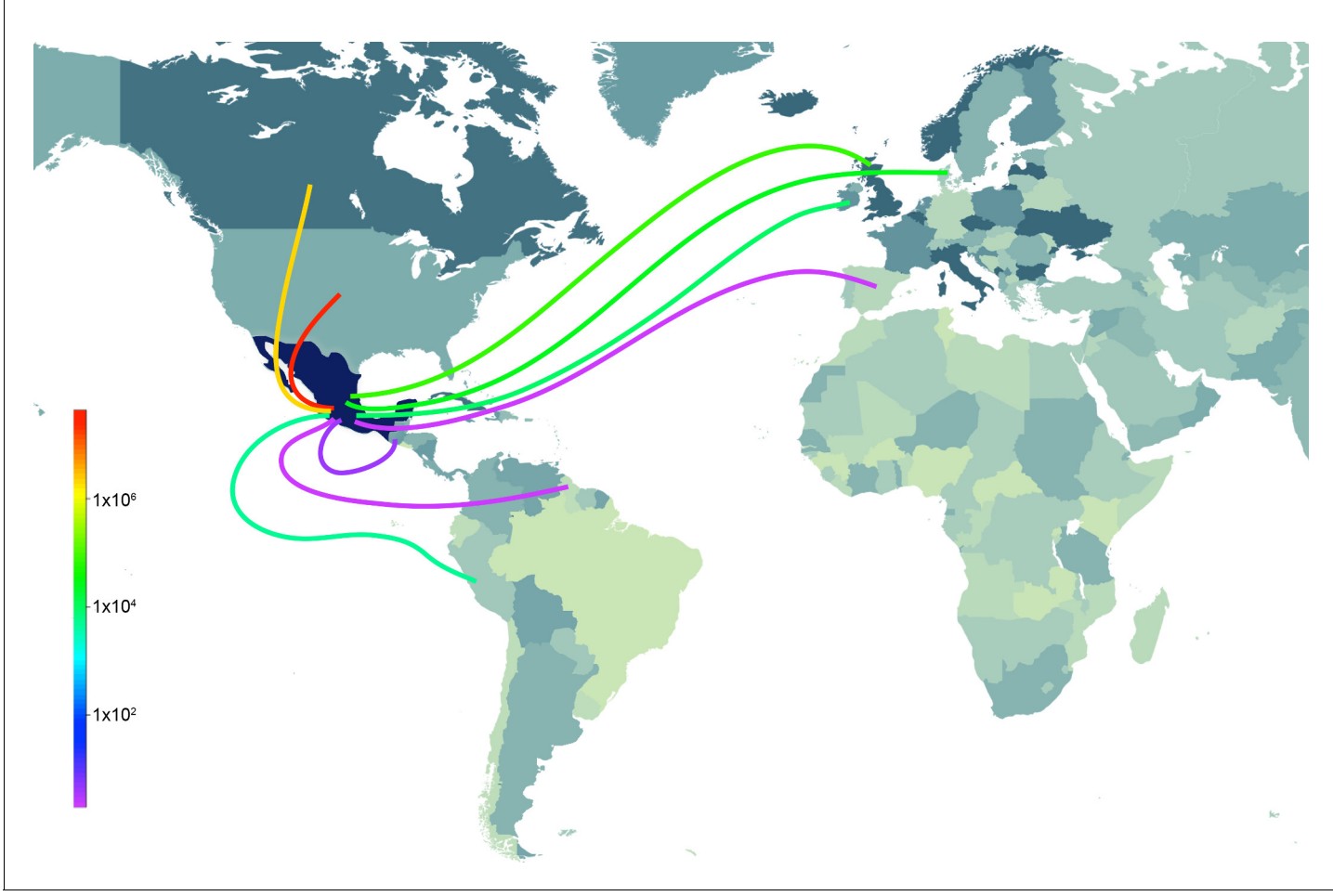

**Figure 6.** Sources of live swine imports into Mexico. Total imports of live swine into Mexico (USD) during 1996–2012 from nine reported trade partners: United States, Canada, Ireland, United Kingdom, Denmark, Spain, Guatemala, Guyana, and Peru. The shade of the line indicates the volume of imports (red = high, purple = low). The shading countries is for purposes of clarity only. Trade data is available from the UN Commodity Statistics Database (**Figure 6—source data 1**).

The following source data is available for figure 6:

**Source data 1.** Pairwise information on imports of live swine from specific countries is available from 1996–2012.

## Spatial dissemination of pdmH1N1 during the early 2009 outbreak in humans

A separate analysis of pdmH1N1 viruses collected globally in humans during the first wave of the 2009 pandemic (March–May 2009) provides additional support for an origin of the pdmH1N1 outbreak in central Mexico, consistent with a previous demonstration of the centrality of Mexico in the 2009 pandemic using epidemiological data (*Brockmann and Helbing, 2013*). The use of genetic data provides more refined within-country spatial details. Location state posterior probabilities for the root of the tree were highest in Mexico City (0.81), consistent with an origin of the pandemic outbreak in Mexico's capital city (*Figure 4—figure supplement 2*). Texas shares a long and heavily-trafficked border with Mexico (*Weinberg et al., 2003*), and the southern United States was the largest recipient of pdmH1N1 introductions from Mexico during the early stages of the 2009 pandemic (*Figure 4B*). However, the southern US was not a major source of pdmH1N1 viruses to the United States' Midwest, West, Northeast regions or globally. Rather, viruses identified in New York, Wisconsin, California, and other US states represent independent introductions of pdmH1N1 from Mexico. Notably, the Northeast region of the United States, including New York, had the largest role in the

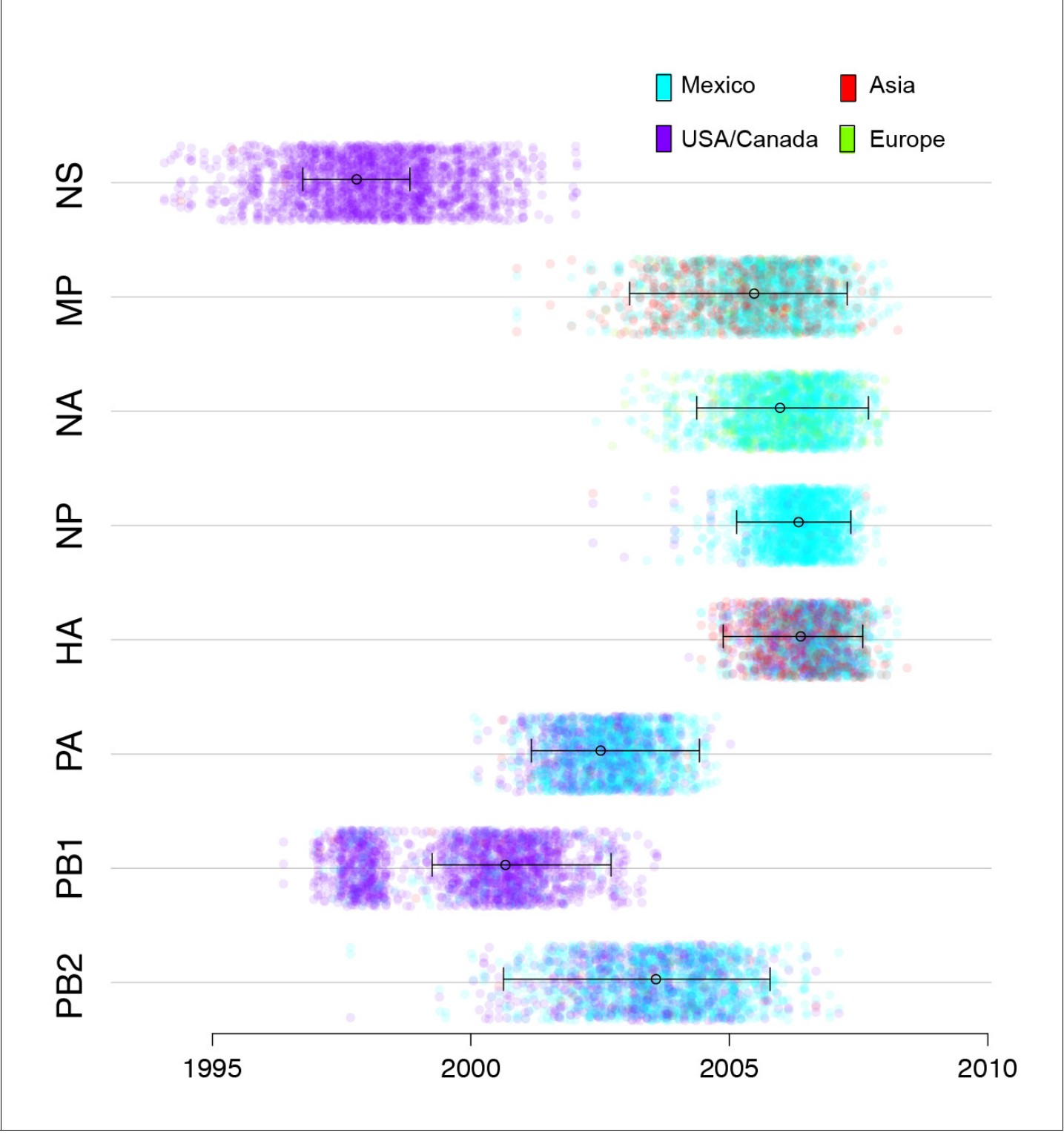

**Figure 7.** Timing and location of swine ancestors of pdmH1N1. The color of each dot represents the inferred location of the node representing the common ancestor of pdmH1N1 viruses and the most closely related swine ivurses (indicated by open circles on the MCC trees in **Figure 3**), for a posterior distribution of ~2000 trees inferred for each segment. A high proportion of blue dots indicates a higher proportion of trees with Mexico as the inferred location state. The x-axis indicates the tMRCA of the same node, again for each tree. The 95% HPD is provided in brackets. Older tMRCAs are associated with longer phylogenetic branch lengths and gaps in sampling.

*Figure 7 continued on next page*

*Figure 7 continued*

The following source data is available for figure 7:

**Source data 1.** Times to the most recent common ancestor (tMRCA) and posterior probabilities (>0.01) for the location state of the node representing the ancestor of the pdmH1N1 clade and the most closely related swine viruses (indicated with open circles in *Figure 3*).

onward global dissemination of the pandemic virus to Europe, Asia, and South America (*Figure 4B*), perhaps driven by high volumes of air traffic from New York City and the size of the city's epidemic during the spring wave (*Viboud et al., 2014*).

## Discussion

Overall, our findings resolve a long-standing question of where the pandemic H1N1 virus originated in swine. Moreover, they underscore the importance of understanding the global evolution of IAVs in pigs, and the pandemic threat presented by the large number of independently evolving swIAV populations worldwide that periodically experience invasions of new viruses. Our findings demonstrate the central importance of inter-hemispheric swine movements in the long-range dissemination of swIAV diversity, which allowed divergent Eurasian and North American viruses to co-circulate in central Mexico and reassort into genotype 1 viruses (*Figure 8*). It remains unclear whether conditions in Mexico were uniquely suitable for the emergence of genotype 1 viruses, or whether it is possible that similar viruses also circulate undetected in Asia, the only other location where Csw, EAsw, and TRsw viruses and various reassortants are known to be present (*Lam et al., 2011*). The inability of researchers to produce genotype 1 viruses experimentally in cell cultures or pigs (*Ma et al., 2014*) implies strain-specific differences in reassortment capabilities. The likelihood of pandemic emergence is therefore enhanced by the seeding of diverse swIAV lineages in countries with varying swine production practices and fitness pressures. The independent evolution of multiple variants increases the probability that one will evolve the capacity to transmit to humans.

Although understanding the positive and negative interactions between fitness in human and swine hosts is central to understanding pandemic emergence, this area of research remains poorly understood. Strong patterns of reassortment between EAsw and North American viruses in Mexican swine are an indication of competitive interactions. The apparent fitness of the TRIG + EAsw MP internal gene constellation in central Mexican swine was central to the emergence of the genotype 1 viruses that gave rise to pdmH1N1, but is not well understood. The success of genotype 1 viruses for many years in central-west Mexico is notable, given the low levels of persistence of pdmH1N1 viruses as an intact genome in swine following reverse zoonosis in many countries (*Liang et al., 2014*; *Nelson and Vincent, 2015*; *Nelson et al., 2015d*; *Pereda et al., 2011*; *Watson et al., 2015*), a pattern also observed in Sonora. The lower levels of onward transmission of pdmH1N1 surface genes in swine could relate to competition with existing H1s and N1s. The identification of EAsw NA and MP segments in central Mexican swine is notable, given that these segments were associated with enhanced respiratory droplet transmission in animal models (*Lakdawala et al., 2011*; *Yen et al., 2011*). The EAsw MP segment also is associated with zoonotic transmission of swIAVs at agricultural fairs in the US (*Bowman et al., 2012*). A number of studies have examined the transmission potential of naturally occurring or lab-generated avian and swine viruses in ferret or other animal models, as a measure of pandemic potential (*Herfst et al., 2012*; *Sorrell et al., 2009*; *Yang et al., 2016*). However, a fundamental understanding of fitness landscapes and trade-offs in multi-host systems has been constrained by the a lack of final resolution on the ethics of various experiments that involve novel viruses that present a potential threat to humans (*Frank et al., 2016*). The identification of viruses related to the 2009 pandemic precursors invites new experimental research into central questions of pandemic emergence at the human-swine interface.

The detection of EAsw in Mexico suggests that even relatively low levels of livestock movement can disseminate a highly transmissible virus such as IAV long distances, bringing attention to the importance of surveillance and mitigation strategies, including quarantine. Presently, imported live swine are not routinely tested for influenza. The emergence of the 2009 pandemic virus was closely linked to the increase in Mexico's imports of live swine during the 1990s and the influx of new influenza virus lineages from the United States and Europe. At this time we cannot rule out Asia-to-

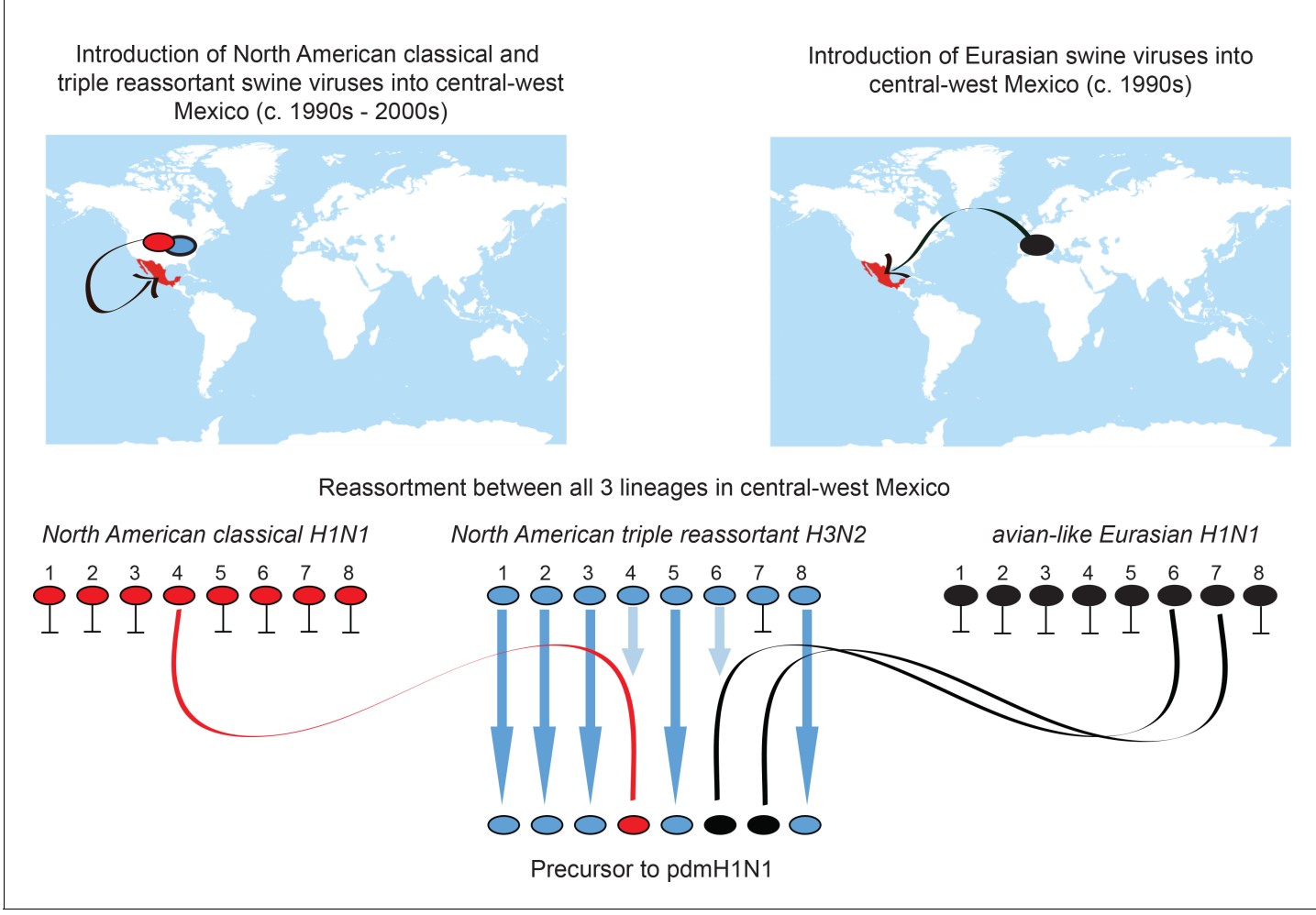

**Figure 8.** Origins of pdmH1N1. Summary of the migration and reassortment events leading to the emergence of pdmH1N1 precursor viruses in central-west Mexican swine. Segments from classical and Eurasian viruses for which there is no evidence of onward transmission in central-west Mexican swine are indicated with short horizontal lines.

Mexico viral migration, owing to possible omissions in reported imports and exports of swine and gaps in sequencing of viruses in swine globally. However, all that is known about the nature of international swine production and trade suggests that this direction of viral movement is unlikely. Long-distance trade of pigs from Europe, the United States, and Canada to countries in Asia and South America occurs primarily to import high-efficiency European and North American breeding lines. Although illegal trade of poultry is important in the spread of AIVs, particularly within the East Asian region, there are no known economic incentives (and strong economic disincentives) for the illegal trade of pigs from Asia and Mexico. Globally, Asia is overwhelmingly a net importer of swine from Europe and North America, with extremely little export of Asian pigs or their viruses to any other continents (*Nelson et al., 2015c*). Another outstanding question is whether genotype 1 viruses have disseminated from Mexico to other countries in Latin America, where surveillance is low or non-existent. Mexico has reported export of live swine to several countries in Latin America (Belize, Colombia, Costa Rica, Cuba, El Salvador). However, the total volume of Mexico's live swine exports during 1996–2012 amounts to only one-tenth of Costa Rica's over the same time period, and one-thirtieth of Brazil's. Mexico is therefore considered an ecological sink for swine influenza viruses, with limited opportunity for viral export to other countries (*Nelson et al., 2015c*).

The unexpected detection of EAsw in Mexico highlights the need to revisit outdated assumptions about the spatial distribution of IAVs in swine populations globally, given the capacity of long-

distance live swine movements to disseminate viral diversity between continents. Although surveillance of swIAVs has been initiated in many new regions since 2009, it remains highly imbalanced on a global scale and low compared to avian surveillance (*Nelson and Vincent, 2015*). Recent detections of novel swIAVs in regions that traditionally have not conducted surveillance in swine, including Latin America and South-east Asia, underscore the importance of expanding surveillance in the large number of under-sampled regions with high densities of swine (*Cappuccio et al., 2011*; *Nelson et al., 2015a*; *Ngo et al., 2012*; *Perera et al., 2013*). China remains an important source of zoonotic influenza viruses with pandemic potential, and the identification of strong regional variations in swIAV diversity in Mexico highlights the importance of capturing swIAV diversity across China's heterogeneous regions. Although the importance of regional live swine movements in the evolution of swIAVs in the United States has been documented (*Nelson et al., 2011*), many countries lack data on the intra-country movements of swine, limiting study of interactions between animal movements and regional patterns of swIAV diversity. Data on poultry movements within and between countries are similarly lacking, limiting our understanding of the role of trade in the spatial dynamics of IAVs in domestic birds. Our ability to predict future pandemics will require intensified viral surveillance and an understanding of how economic forces and international trade policies affect changes in animal movements and production practices that drive viral emergence.

## Materials and methods

### Collection of influenza viruses in Mexican swine

Swine influenza isolates were obtained from samples submitted to Avimex laboratories (Mexico) for confirmation diagnostics of pigs with respiratory syndrome. The samples covered years 2010 (1 sample), 2011 (6 samples), 2012 (34 samples), 2013 (14 samples) and 2014 (3 samples). All samples came from large commercial facilities from 6 states: Sonora (8 different farms), Jalisco (6 farms), Guanajuato (3 farms), Puebla (2 farms), Yucatan (2 farms) and Aguas Calientes (1 farm). Virus was isolated in specific pathogen free embryonated eggs or in cell culture; HA, NA subtype was determined and samples from first or second passage were sent to Mount Sinai (New York) for RNA extraction and whole genome sequencing. When multiple isolates of the same subtype were obtained from the same farm and outbreak the sample with highest viral titer was submitted. GenBank accession numbers of the 58 genomes are available in the Dryad data repository under Doi:10.5061/dryad.m550m. *Figure 2—source data 1* describes the date, location, subtype and lineage assignment of each viral segment.

### Influenza A genome sequencing

Viral RNA was obtained from 140 µl of the original sample, or after amplification in MDCK cells, using the QIAamp Viral RNA minikit QIAGEN). Next, 5 µl of viral RNA were used as template in a 50 µl multisegment RT-PCR reaction (Superscript III high-fidelity RT-PCR kit) with influenza-specific universal primers complementary to the conserved 12–13 nucleotides at the end of all 8 genomic segments. Primer sequences and final concentrations in the reaction were as follows (influenza-complementary sequences are underlined): Opti1-F1 – 5' GTTACGCGCCAGCAAAAGCAGG (0.1 µM); Opti1-F2 - 5' G TTACGCGCCAGCGAAAGCAGG (0.1 µM); Opti1-R1 - 5' GTTACGCGCCAGTAGAAACAAGG (0.2 µM). Amplicons were purified with 0.45x volume AMPure XP beads (Beckman Coulter) and 0.5–1 µg was sheared to an average fragment size of 150 bp on a Bioruptor Pico sonicator (Diagenode). Next, amplicon sequence libraries were prepared using the end repair, A-tailing, and adaptor ligation NEBNext DNA library prep modules for Illumina from New England Biolabs according to the manufacturer's protocol. Following final size-selection with 1x volume Ampure XP beads, and secondary PCR (8 cycles) to introduce barcoded primers, multiplexed libraries were sequenced on the Illumina HiSeq 2500 platform in a single-end 100 nt run format.

### De novo genome assembly

Single-end 100 nt reads were first filtered with cutadapt (*Martin, 2011*) to remove low-quality sequences and adapters. Reads were then mapped against a non-redundant copy of IAV sequences in the Influenza Research Database (IRD) using STAR (*Dobin et al., 2013*) and chimeric reads with non-contiguous alignments to reference segments (typically originating from defective interfering

particles containing segments with internal deletions) were removed. Assembly of IAV genomic segments was performed using a custom pipeline (available from the authors) in multiple stages. First, an initial assembly was done using the inchworm component of Trinity (*Grabherr et al., 2011*), and viral contigs bearing internal deletions were identified by BLAT (*Kent, 2002*) mapping against non-redundant IRD reference sequences. In the second stage the inchworm assembly was repeated but now removing breakpoint-spanning kmers from the assembly graph. Resulting IAV contigs were then oriented, and trimmed to remove low-coverage ends and any extraneous sequences beyond the conserved IAV termini. In the final stage the CAP3 (*Huang and Madan, 1999*) assembler was used to improve contiguity by merging contigs originating from the same segment type if their ends overlapped by at least 25 nt. Assembly quality and contiguity was assessed for all segments by mapping sequence reads back to the final assemblies using the Burrows-Wheeler Alignment (BWA) tool (*Li and Durbin, 2009*) and complete segments were annotated using the NCBI Influenza Virus Sequence Annotation Tool (*Bao et al., 2007*).

## Phylogenetic analysis of swIAVs in Mexico

In addition to the 58 whole-genome sequences generated for this study from Mexico, background data sets were included that were recently used in previous studies of IAV evolution in swine (*Nelson et al., 2014*, *2015a*, *2015b*, *2015c*). The vast majority of swIAV sequences from the Americas are deposited in NCBI's GenBank, but to ensure that no additional sequences of importance were available from the Global Initiative on Sharing All Influenza Data's (GISAID) Epiflu database from Europe and Asia, sequences for the EAsw lineage were downloaded from GISAID (acknowledgement of the authors and research laboratories that contributed these data are provided in *Supplementary files 1–2*). A large number of whole-genome sequences from swIAVs in European swine were published by *Watson et al. (2015)* after we completed our analysis. To ensure that none were closely related to pdmH1N1 or any of the Mexican swIAVs, particularly on the trees inferred for the Eurasian N1 and MP segments, we inferred neighbor-joining trees including the new European swIAV sequences deposited into GenBank by these authors (data available from the authors). Sequence alignments were constructed for each of the six internal gene segments (PB2, PB1, PA, NP, MP, and NS) and for the H1, H3, N1, and N2 antigenic segments separately using MUSCLE v3.8.3 (*Edgar, 2004*), with manual correction in Se-Al v2.0 (available at http://tree.bio.ed.ac.uk/software/seal/). The program Path-O-Gen v1.4.0 (available at http://tree.bio.ed.ac.uk/software/pathogen/) was used to identify potential sequencing errors that substantially deviated from the linear regression of root-to-tip genetic distance against time, which were subsequently removed from the study. We excluded nine triple reassortant Mexican swIAVs (e.g., A/swine/Mexico/Mex19/2010 (H1N1), *Figure 3—figure supplement 1*) that had shorter branch lengths than expected given their year of sampling, an indication of possible errors in sequencing or assembly.

For the purposes of lineage assignment, initial phylogenetic trees were inferred using the neighbor-joining method available in PAUP v4.0b10 for each of the ten alignments (available at http://paup.csit.fsu.edu). For the PB2, PB1, PA, NP, and NS segments, each virus was categorized as related to the triple reassortant internal genes ('TRIG') lineage, classical lineage (for NP and MP), avian-like Eurasian (MP), or pdmH1N1. Reference sequences for pdmH1N1 from humans and swine, including A/California/04/2009(H1N1), were included to assist identification of the pdmH1N1 lineage. H1 and N1 segments were categorized as (a) classical, (b) avian-origin Eurasian, (c) pdmH1N1, or (d) human seasonal H1N1 virus origin. All H3 and N2 segments belonged to the same lineage: human seasonal H3N2 virus-related. The sequence alignment for each segment was further separated into each of these lineages, and trees were inferred independently for each alignment that contained at least one Mexican virus: PB2-TRIG (n = 273 sequences), PB1-TRIG (n = 329 sequences), PA-TRIG (n = 309 sequences), H1-classical (n = 352 sequences), H1-human seasonal (n = 213 sequences), H1-eurasian (n = 305 sequences), H3 (n = 331 sequences), NP- TRIG/classical (n = 291 sequences), N1-classical (n = 287 sequences), N1-Eurasian (n = 314 sequences), N2 (n = 726 sequences), MP-TRIG/classical (n = 256 sequences), MP-Eurasian (n = 300 sequences), and NS-TRIG/classical (n = 479 sequences). For the NP, MP, and NS segments, the TRIG lineage is a continuation of the classical lineage, as classical NP, MP, and NS segments were incorporated into triple reassortant viruses during the reassortment (*Zhou et al., 1999*).

Phylogenetic relationships were inferred for each of the data sets separately using the time-scaled Bayesian approach using MCMC available via the BEAST v1.8.2 package (*Drummond et al., 2012*)

and the high-performance computational capabilities of the NIH HPC Biowulf Linux cluster at the National Institutes of Health, Bethesda, MD (http://hpc.nih.gov/). A relaxed uncorrelated lognormal (UCLN) molecular clock was used, with a constant population size, and a general-time reversible (GTR) model of nucleotide substitution with gamma-distributed rate variation among sites. For viruses for which only the year of viral collection was available, the lack of tip date precision was accommodated by sampling uniformly across a one-year window from January 1st to December 31st. The MCMC chain was run separately at least three times for each of the data sets and for at least 100 million iterations with sub-sampling every 10,000 iterations, using the BEAGLE library to improve computational performance (*Suchard and Rambaut, 2009*). All parameters reached convergence, as assessed visually using Tracer v.1.6, with statistical uncertainty reflected in values of the 95% highest posterior density (HPD). At least 10% of the chain was removed as burn-in, and runs for the same lineage and segment were combined using LogCombiner v1.8.0 and downsampled to generate a final posterior distribution of 1000 trees that was used in subsequent analyses.

In order to visualize branch lengths between swIAVs and the pdmH1N1 clade in terms of genetic distance, rather than time, the phylogenetic relationships of each of the eight data sets containing the pdmH1N1 clade were inferred using the maximum likelihood (ML) method available in the program RAxML v7.2.6 (*Stamatakis, 2006*), incorporating a general time-reversible (GTR) model of nucleotide substitution with a gamma-distributed (Γ) rate variation among sites. To assess the robustness of each node, a bootstrap resampling process was performed (500 replicates), again using the ML method available in RAxML v7.2.6. Although not the focus of this study, phylogenetic trees also were inferred for the pdmH1N1 lineage to identify recent human-to-swine transmission events in Mexico (inferred for PB2 and H1, as representative segments, *Figure 3—source data 4*), using methods described above and background data sets used in previous studies (*Nelson et al., 2015b*; *2015d*).

## Spatial analysis of swIAVs in Mexico

The phylogeographic analysis considered 11 locations: Asia (including China, Japan, South Korea, Thailand, Vietnam), USA/Canada, Europe (including Belgium, Czech Republic, Denmark, France, Germany, Italy, Netherlands, Poland, Spain, and the United Kingdom), South America (including Argentina and Brazil), Mexico (Aguascalientes), Mexico (Guanajuato), Mexico (Jalisco), Mexico (Puebla), Mexico (Sonora), and Mexico (Yucatan), as well as humans (globally). The location state was specified for each viral sequence, allowing the expected number of location state transitions in the ancestral history conditional on the data observed at the tree tips to be estimated using 'Markov jump' counts (*Minin and Suchard, 2008a*), which provided a quantitative measure of asymmetry in gene flow between regions. The location of viruses in the pdmH1N1 clade was left uninformed, allowing the reconstruction of the location state of the common ancestor of pdmH1N1 to be unbiased by human pdmH1N1 data. For computational efficiency the phylogeographic analysis was run using an empirical distribution of 1000 trees (*Lemey et al., 2014*), allowing the MCMC chain to be run for 25 million iterations, sampling every 1000. A Bayesian stochastic search variable selection (BSSVS) was employed to improve statistical efficiency for all data sets containing more than four location states. Maximum clade credibility (MCC) trees were summarized using TreeAnnotator v1.8.0 and the trees were visualized in FigTree v1.4.2. Heat-maps were constructed using the R package to summarize Markov jump counts inferred over the totality of phylogenies (all segments, all swIAV lineages). Waiting times between location state transitions ('Markov rewards') (*Minin and Suchard, 2008b*) also were estimated, allowing us to estimate 'Markov jump' counts normalized for the percent time of locations in the tree.

## Spatial analysis of human pdmH1N1

All available whole-genome sequence data for pdmH1N1 viruses collected globally in humans during March 1–May 31, 2009 were downloaded from the National Center for Biotechnology Information's Influenza Virus Resource (*Bao et al., 2008*) (http://www.ncbi.nlm.nih.gov/genomes/FLU/FLU.html) on December 12, 2015. The over-sampled locations of New York, USA and Wisconsin, USA were randomly subsampled to 50 genomes each, resulting in a final data set of 422 complete concatenated whole-genome sequences (*Supplementary file 3*). The sequence data were aligned using MUSCLE v3.8.3 (*Edgar, 2004*). Phylogenetic relationships were inferred using the time-scaled Bayesian

approach using MCMC available via the BEAST v1.8.2 package, in this case assuming a model of exponential growth in the number of infections, as this model is more realistic during the early growth stage of new pandemic virus (*Smith et al., 2009*). The alignment and phylogeographic analysis, using the methods described above, considered 12 locations: Asia (China, Japan, South Korea, Malaysia, Singapore, Thailand), Canada, Europe (United Kingdom, Finland, Italy, Russia, Netherlands, France, Germany), South America (Argentina, Chile, as well as the Dominican Republic), Mexico (Mexico City), Mexico (San Luis Potosi), Mexico (Veracruz), Mexico (state unknown), United States - Northeast (Maryland, Massachusetts, New Jersey, New York, Rhode Island), United States – Midwest (Indiana, Iowa, Kansas, Michigan, Nebraska, Ohio, Wisconsin), United States – South (Alabama, Texas, Florida, Mississippi, South Carolina, Tennessee), and United States – West (Arizona, California, Colorado, New Mexico). Heat-maps were constructed using the R package to summarize Markov jump counts inferred over the phylogeny. Waiting times between location state transitions ('Markov rewards') (*Minin and Suchard, 2008b*) also were estimated, allowing us to estimate 'Markov jump' counts normalized for the percent time of locations in the tree.

## Genotype 1

To estimate the extent of reassortment and onward transmission of different genotypes in swine in Mexico, we used a similar phylogeographic approach, but in this case specified a location state for each Mexican virus based on genotype. For viruses collected for this study, we used the 12 genotypes defined in *Figure 2*. For the six additional whole-genome sequences available from Mexico that were published previously (*Nelson et al., 2015c*) we defined two new genotypes: genotypes 13 and 14. In addition to estimating transitions between locations (Markov jumps), we also estimated the waiting times between transitions ('Markov rewards') (*Minin and Suchard, 2008b*) as an indication of the period of time Mexican lineages circulate as different genotypes.

## Live swine import into Mexico

The trade value (USD) for live swine trade between other countries and Mexico for the years 1996– 2012 was obtained from the United Nations' Commodity Trade Statistics Database (available at http://comtrade.un.org, accessed March 20, 2014) (*Figure 6—source data 1*). Estimates of the total number of live swine imports into Mexico (country of origin unknown) were obtained for a longer time period (1969–2010) from the Food and Agriculture Organization (FAO) of the United Nations Datasets repository (available at http://data.fao.org/datasets, accessed March 21, 2014) (*Figure 5— source data 1*). Data on the size of live swine population in each of Mexico's states was obtained from Mexico's Secretariat of Agriculture, Livestock, Rural Development, Fisheries and Food (SAGARPA) (www.siap.gob.mx), accessed December 30, 2015 (*Figure 1—source data 1*).

## Acknowledgements

We would like to thank the research groups that contributed sequence data to the Epiflu data base at GISAID and NCBI's Influenza Virus Resource at GenBank, with particular thanks extended to Rebecca Halpin at the J Craig Venter Institute, Rockville, MD, for providing additional spatial information for viruses collected in Mexico in humans. Robert Gaffey, a Research Assistant at the Fogarty International Center, provided assistance in organizing supplementary information. We also want to thank Richard Cadagan for excellent technical assistance.

## Additional information

### Funding

| Funder | Grant reference number | Author |
|---|---|---|
| National Institutes of Health | Centers of Excellence for Influenza Research and Surveillance, Contract No. HHSN272201400008C | Adolfo García-Sastre |
| Fogarty International Center | Multinational Influenza Seasonal Mortality Study | Martha I Nelson |

The funders had no role in study design, data collection and interpretation, or the decision to submit the work for publication.

## Author contributions

IM, MIN, Acquisition of data, Analysis and interpretation of data, Drafting or revising the article; FQ-M, RC-F, JHL-P, FC-P, BL-D, Collected and processed samples, Contributed unpublished essential data or reagents; JD, LFC, NST, HvB, Acquisition of data, Analysis and interpretation of data; AR, Analysis and interpretation of data, Drafting or revising the article; AG-S, Conception and design, Analysis and interpretation of data, Drafting or revising the article

## Author ORCIDs

Nídia S Trovão, http://orcid.org/0000-0002-2106-1166
Adolfo García-Sastre, http://orcid.org/0000-0002-6551-1827

# Additional files

## Supplementary files

• Supplementary file 1. Dataset 1.

• Supplementary file 2. Dataset 2.

• Supplementary file 3. Dataset 3.

• Reporting Standard 1. The sample and sequence data were collected according to the following CEIRS DPCC standard: CEIRS DPCC Sequence Submission FASTA Reference v2.0

• Reporting Standard 2. The sample and sequence data were collected according to the following CEIRS DPCC standard: CEIRS DPCC Sequence Submission Metadata Reference v2.0

## Major datasets

The following datasets were generated:

| Author(s) | Year | Dataset title | Dataset URL | Database, license, and accessibility information |
|---|---|---|---|---|
| Martha I Nelson, Francisco Quezada-Monroy, Jayeeta Dutta, Refugio Cortes-Fernández, J Horacio Lara-Puente, Felipa Castro-Peralta, Luis F Cunha, Nídia Sequeira-Trovão, Bernardo Lozano-Dubernard, Andrew Rambaut, Harm van Bakel, Adolfo García-Sastre, Ignacio Mena | 2016 | Swine influenza A viruses collected in Mexico 2010-2014 | http://www.ncbi.nlm.nih.gov/bioproject/?term=PRJNA315383 | Publicly available at NCBI BioProject (Accession no: PRJNA315383) |

Ignacio Mena, Martha I Nelson, Francisco Quezada-Monroy, Jayeeta Dutta, Refugio Cortes-Fernández, J Horacio Lara-Puente, Felipa Castro-Peralta, Luis F Cunha, Nídia Sequeira-Trovão, Bernardo Lozano-Dubernard, Andrew Rambaut, Harm van Bakel, Adolfo García-Sastre | 2016 | Data from: Origins of the 2009 H1N1 influenza pandemic in swine in Mexico | http://dx.doi.org/10.5061/dryad.m550m | Available at Dryad Digital Repository under a CC0 Public Domain Dedication

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
