## [Decision Letter]

Thank you for submitting your article "Origins of the 2009 H1N1 influenza pandemic in swine in Mexico" for consideration by *eLife*. Your article has been reviewed by three peer reviewers, one of whom is a member of our Board of Reviewing Editors and the evaluation has been overseen by Arup Chakraborty as the Senior Editor. The reviewers have opted to remain anonymous.

The reviewers have discussed the reviews with one another and the Reviewing Editor has drafted this decision to help you prepare a revised submission.

Summary:

The manuscript presents a phylogenetic analysis of 58 full influenza A virus genomes collected from swine in Mexico to shed light on the origin of the 2009 pandemic of H1N1 influenza. The newly sequenced genomes close important gaps in our understanding of the emergence of this pandemic. The authors show that virus variants with sequences closely related to all segments of human H1N1pdm09 circulate in Mexican swine. This finding singles out Mexico as the likely region where the pandemic virus emerged. These results emphasize the need for concerted global surveillance of influenza virus diversity in animals.

All reviewers agree that the data shed light on an important and interesting problem. However, before we can recommend acceptance, a number of issues should be addressed.

Essential revisions:

1) More details on sample collection: How many samples were tested in different years? How were animals chosen for testing? Did these animals come from large commercial facilities or backyard farms? What were the criteria for sequencing positive samples? Within each location (Sonora, Puebla, etc) did the samples consistently come from the same producers/farms?

2) We would like to see a more extensive discussion of the gaps in surveillance in swine (not only in Mexico, but also Asia and Europe), and how these gaps limit the inferences that can be made.

---

## [Author Response]

Essential revisions:

1) More details on sample collection: How many samples were tested in different years? How were animals chosen for testing? Did these animals come from large commercial facilities or backyard farms? What were the criteria for sequencing positive samples? Within each location (Sonora, Puebla, etc) did the samples consistently come from the same producers/farms?

Following the reviewers’ suggestions, we have extended the description of the sample collection and virus isolation in the Materials and Materials and methods section to include all the information requested by the reviewers. The revised paragraph is:

“Collection of influenza viruses in Mexican swine. Swine influenza isolates were obtained from samples submitted to Avimex laboratories (Mexico) for routine confirmation diagnostics of pigs with respiratory syndrome. […] GenBank accession numbers of the 58 genomes are available in the Dryad data repository under DOI: 10.5061/dryad.m550m. Figure 2—figure supplement 1 describes the date, location, subtype and lineage assignment of each viral segment.”

2) We would like to see a more extensive discussion of the gaps in surveillance in swine (not only in Mexico, but also Asia and Europe), and how these gaps limit the inferences that can be made.

This is a very important issue, and our revised manuscript addressed this matter in further detail in the Results and the Discussion. We have also changed many of our references from ‘European swine’ to ‘Eurasian swine’ to better reflect ambiguities about the source Eurasian viruses in Mexico.

Results; “We also recognize the need to draw phylogeographic inferences with an awareness of missing data. Major gaps in surveillance in swine in Europe and Asia during the 1990s have resulted in long branch lengths between swIAVs collected in Mexico and Eurasia (Figure 3—figure supplement 1). Although it is difficult to infer from the tree whether the EAsw MP identified in Mexico was introduced from Asian or European swine (Figure 4—figure supplement 1), the lack of live swine imports reported between Mexico and any Asian country (Figure 6), and the low export of Asian swine globally (Nelson et al., 2015c) suggest that Europe may be a more likely source of the Mexican viruses, similar to the H1 and N1 segments.”

Discussion; “At this time we cannot rule out Asia-to-Mexico viral migration, owing to possible omissions in reported imports and exports of swine and gaps in sequencing of viruses in swine globally. […] Globally, Asia is overwhelmingly a net importer of swine from Europe and North America, with extremely little export of Asian pigs or their viruses to any other continents (Nelson et al., 2015c).”